**Data Availability Statement:** Access to data and Data sharing: CZ have full access to all the data used in the study and takes responsibility for the integrity of the data and the accuracy of the data

# Utilisation and costs of mental health-related service use among adolescents

**Carolina Ziebold** [1], **Wagner Silva-Ribeiro**[1,2], **Derek King**[2], **David McDaid** [2], **Mauricio Scopel Hoffmann**[2,3,4,5], **Renee Romeo**[6], **Pedro Mario Pan**[1,5], **Eurípedes Constantino Miguel**[5,7], **Rodrigo Affonseca Bressan**[1,5], **Luis Augusto Rohde**[5,8,9], **Giovanni Abrahão Salum**[5,9], **Jair de Jesus Mari**[1,5], **Sara Evans-Lacko** [2]*

1 Departamento de Psiquiatria, Universidade Federal de São Paulo, São Paulo, Brazil, 2 Care Policy and Evaluation Centre, London School of Economics and Political Science, London, United Kingdom, 3 Universidade Federal de Santa Maria, Santa Maria, Brazil, 4 Hospital de Clínicas de Porto Alegre, Porto Alegre, Brazil, 5 National Institute of Developmental Psychiatry for Children and Adolescents, São Paulo, Brazil, 6 King's College London, London, United Kingdom, 7 Universidade de São Paulo, São Paulo, Brazil, 8 ADHD Outpatient Program & Developmental Psychiatry Program, Hospital de Clínicas de Porto Alegre, Brazil, 9 Universidade Federal do Rio Grande do Sul, Porto Alegre, Brazil

* S.Evans-Lacko@lse.ac.uk

## Abstract

### Background

The high level of care needs for adolescents with mental health conditions represents a challenge to the public sector, especially in low and middle-income countries. We estimated the costs to the public purse of health, education, criminal justice and social care service use associated with psychiatric conditions among adolescents in Brazil; and examined whether the trajectory of psychopathology and its impact on daily life, and parental stigma towards mental illness, was associated with service utilisation and costs.

### Methods

Data on reported service use among adolescents from a prospective community cohort (n = 1,400) were combined with Brazilian unit costs. Logistic regression and generalised linear models were used to examine factors associated with service use and associated costs, respectively.

### Results

Twenty-two percent of those who presented with a psychiatric disorder used some type of service for their mental health in the previous twelve months. Higher odds of service use were associated with having a diagnosed mental disorder (either incident, [OR = 2.49, 95% CI = 1.44–4.30, p = 0.001], remittent [OR = 2.16, 95%CI = 1.27–3.69, p = 0.005] or persistent [OR = 3.01, 95%CI = 1.69–5.36, p<0.001]), higher impact of symptoms on adolescent's life (OR = 1.32, 95%CI = 1.19–1.47, p<0.001) and lower parental stigma toward mental illness (OR = 1.12, 95%CI = 1.05–1.20, p = 0.001). Average annual cost of service use was 527.14 USD (s.d. = 908.10). Higher cost was associated with higher disorder impact (β =

analysis. Data were provided by the Brazilian High-Risk Cohort study and are available upon request in the Open Science Framework public repository (https://osf.io/ktz5h/).

**Funding:** The research presented in this article was funded by the European Research Council under the European Union's Seventh Framework Programme (FP7/2007-2013)/ERC grant agreement no 337673, and supported by the UK Medical Research Council (MR/R022763/1), National Institute of Developmental Psychiatry for Children and Adolescents, a science and technology institute funded by Conselho Nacional de Desenvolvimento Científico e Tecnológico (CNPq; National Council for Scientific and Technological Development; grant numbers 573974/2008-0 and 465550/2014-2) and Fundação de Amparo à Pesquisa do Estado de São Paulo (FAPESP grant number 2008/57896-8 and 2014/50917-0). SEL receives support from the UK Medical Research Council in relation to the Mentalkit-Brazil project (MR/R022763/1) and the Economic and Social Research Council. CZ received a doctoral scholarship and research abroad scholarship by the Fundação de Amparo à Pesquisa do Estado de São Paulo (grant number 2018/05586-7 and 2019/08731-0). MSH was supported by the Newton International Fellowship (Ref: NIF\R1\181942), awarded by the Academy of Medical Sciences through the UK Government's Newton Fund Programme. The funding organisations had no role in the study design; collection, analysis or interpretation of data; in the writing of this article; or in the decision to submit the article for publication.

**Competing interests:** All authors report no conflict of interest associated with this publication. Luis Augusto Rohde has received grant or research support from, served as a consultant to, and served on the speakers' bureau of Aché, Bial, Medice, Novartis/Sandoz, Pfizer/Upjohn, and Shire/Takeda in the last three years. The ADHD and Juvenile Bipolar Disorder Outpatient Programs chaired by Dr Rohde have received unrestricted educational and research support from the following pharmaceutical companies in the last three years: Novartis/Sandoz and Shire/Takeda. Dr Rohde has received authorship royalties from Oxford Press and ArtMed. None of these commercial relationships alter our adherence to PLOS ONE policies on sharing data and materials.

0.25, 95%CI = 0.12–0.39, p<0.001), lower parental stigma (β = 0.12, 95%CI = 0.02–0.23, p = 0.020) and white ethnicity (β = 0.55, 95%CI = 0.04–1.07, p = 0.036).

## Conclusion

The impact of mental health problems on adolescents' daily lives and parental stigmatising attitudes toward mental illness were the main predictors of both service use and costs.

## Introduction

Mental health conditions affect 13.4% of children and adolescents globally, representing the leading cause of disability in this age group [1]. They can have long-term impacts on health and social outcomes into adulthood [2–7]. The high prevalence and potentially enduring nature of these impacts make addressing youth mental health conditions particularly important, but this is a challenge for public systems with limited resources [8]. Economic costs associated with youth mental health conditions involve a wide range of sectors including health, educational, social care, and criminal justice services [9, 10]. This can represent a substantial cost to the public purse, yet it could also be considered a wise investment given the evidence that effective treatment can mitigate the impact of poor mental health [2]. Estimating the economic cost of mental disorders in young people from the perspective of the public purse and understanding which factors are associated with these costs could support more effective and efficient policy planning and care delivery [8, 11, 12].

Some studies from high–income countries about mental health-related service use among young people suggest that lower socioeconomic status, as well as clinical features (illness severity and impact of disorders) increase the likelihood of service usage in the health, special education, and social care sectors, while male gender and older age are associated with more criminal justice services contacts [11, 13, 14]. These sociodemographic and clinical characteristics are also associated with greater mental health-related treatment costs among young people [11, 14, 15]. Stigma can also influence help-seeking. Research in adult populations show clear links between stigma and reduced help-seeking [16], reduced adherence to treatment and early withdrawal from services [16, 17]. Additionally, stigma is one of the most reported barriers to help-seeking amongst adolescents [18]. Families also play a central role in young people's contact with services. However, there is limited research about how parental stigma could impact on service use and costs. Stigmatising attitudes toward mental illness amongst parents may influence service contacts due to shame and fear of labelling their youth's mental health condition [19]. We know of two papers from the UK (using the same data), where lower mental illness-related stigma among caregivers (n = 407), was associated with an increased likelihood of young people's mental health service use [19], though it was not associated with costs [20]. We know of no studies on the association between mental illness-related stigma among caregivers and young people's service use and costs outside the UK or in low and middle-income countries (LMICs), where families and young people may face different types of barriers to mental health care. Additionally, most costing studies have focused on a single disorder, commonly autism, attention deficit hyperactivity disorder or conduct disorders [10], and little is known about how, in addition to the type of disorder, whether persistence of psychopathology from childhood to adolescence, disorders' impact on adolescent's daily life (i.e. functioning), and key barriers to care such as stigma, could influence costs.

There are a limited number of studies reporting on prevalence of mental health service use in LMICs [21–23], however, none use validated service use measures. Moreover, prevalence of any use does not capture the intensity of use (e.g. number or type of visits) needed to understand the economic impact of child mental health problems. From a global mental health perspective, examining this issue in a LMIC context, where resources are scarce, is of major significance. Brazil provides universal access to health services and education for the entire population that is free at the point of use, while private health care and education are used by about 20% of the population [24–26]. Estimating the economic cost of mental disorders among young people to the public purse, and understanding which factors are associated with these costs in Brazil is essential for public policy planning, specifically to optimise investment. This approach could also be of value for similar health and welfare systems. Furthermore, examining the variation in costs according to clinical characteristics of adolescents, beyond type of diagnosis, is important as the impact of psychopathology on daily life and the trajectory of psychopathology from childhood to adolescence, may support service planning and resource allocation in relation to clinical characteristics in a preventive and responsive way.

The aim of this study is to estimate the costs associated with health, education, criminal justice and social care services among a cohort of young people in Brazil. We first present the annual aggregate cost to the public purse and then disaggregate this impact to reflect and understand the relative costs to different sectors. Second, we examine how costs vary according to: mental health trajectories, impact of the disorder on everyday life, and parent/guardian stigma towards mental illness. We hypothesise that persistence of psychiatric disorders from childhood to adolescence and associated impact on adolescents' lives have the greatest influence on costs. However, we also expect that lower levels of parental stigma towards mental illness will predict greater likelihood of service use and hence higher costs.

## Methods

### Data and participants

This study is nested within the Brazilian High-Risk Cohort (BHRC), which is an ongoing prospective longitudinal study that comprises a community sample and a high-risk sub-sample (a sample at increased risk of mental disorders) of young people from Sao Paulo and Porto Alegre, Brazil. A detailed description of the sample and procedures can be found elsewhere [27]. Briefly, during the registry day, 12,500 parents of young people aged 6 to 14 years attending 57 schools (22 in Porto Alegre and 35 in São Paulo) were invited to a screening of mental health disorders using the Family History Screen (FHS) [28]. A total of 8,012 families (9,937 eligible children, 45,394 family members) were interviewed. Based on the percentage of members in the family that screened positively for psychiatric disorders, an index of family load for each potential eligible child was computed. The final cohort comprised 2,511 young people; 957 were randomly selected, and 1,554 were a sub-sample at increased risk of mental disorders based on the FHS. Cohort participants were interviewed at baseline (aged 6–14 years, calendar year:2010–2011, n = 2,511), and at first follow-up (n = 2010, aged 9–17 years, calendar year 2014, 80% retention rate). Due to an administrative error, the service use questions were not included in the interview schedule for the first 129 participants (6%) of the BHRC first follow-up. Therefore, we were only able to contact a subsample of those who participated at the first follow-up (94%, n = 1,881) to respond to a supplementary interview which included a comprehensive assessment of mental health-related service use (calendar year: 2014–2015, young people participants aged 10–18 years). Among those contacted, 1,400 (74.4%) guardians (in 93.1% of cases the biological mother) completed the interview, 982 (70.1%) by telephone and 418 (29.9%) face-to-face (See flow chart in S1 Fig). There were no significant differences in

persistence of psychopathology or impact of psychopathology on adolescents' lives among respondents versus non-respondents.

This research was carried out in accordance with the latest version of the Declaration of Helsinki. Parental written informed consent was obtained from all the research subjects. Young people provided verbally informed assent (documented as part of the consent form, and witnessed by the interviewer), and those who were able to read and write also provided written consent. All procedures were approved by the Ethics Committee of the Federal University of São Paulo-UNIFESP (N° 2.879.533 and CAAE 06457219.9.0000.5505), Hospital de Clínicas de Porto Alegre (CAAE 06457219.9.3001.5327) and the European Research Commission. Data were provided by the Brazilian High-Risk Cohort study and are available upon request in the Open Science Framework public repository (https://osf.io/ktz5h/).

### Measures

**Sociodemographic characteristics.** Data on the following sociodemographic characteristics were collected: gender, age at follow-up, ethnicity (white and non-white: black, Asian, indigenous or mixed-race), socioeconomic group (SEG), and maternal educational level (no/basic, secondary or university education). SEG was defined according to a Brazilian standardized questionnaire [29]. Based on families' assets and head of household's education level, a total score ranging for 0 to 46 is given, where greater scores represent higher socioeconomic status. In this study, SEG was categorised as "low" (0–22) and "high" (23–46).

**Psychopathology.** *Psychiatric diagnosis.* Psychiatric diagnoses were assessed at baseline and follow-up using the Brazilian-Portuguese version of the Development and Well-being Assessment (DAWBA) [30, 31], which is a highly structured interview used to generate DSM-IV diagnoses. Trained interviewers gathered information on current problems causing significant distress or social impairment. At baseline, diagnostic assessment and interviews were performed with guardians only. Previous literature has found that self-report on internalising conditions during adolescence is higher compared with parental report. This can be explained because internalising problems, such as anxiety or depression, would be less observable by guardians, being advisable to consider both reports to reach a reliable evaluation of adolescent mental health [32, 33]. For this reason, diagnostic assessment at 3-year follow-up was performed considering guardian reports and additional information from interviews with the young people about internalising conditions. Computerised diagnostic probabilities were then generated based on responses those were carefully evaluated by 9 trained psychiatrists who determined the diagnosis.

*Broad psychiatric diagnostic categories.* Based on previous literature [34], follow-up DAWBA diagnoses were grouped into three broad categories: distress-related disorders (including depression, generalised anxiety disorder, obsessive–compulsive disorder, tic, eating disorder), fear-related disorders (including panic, agoraphobia, social anxiety, specific phobia and separation anxiety) and externalising disorders (including conduct disorder, oppositional defiant disorder and attention deficit/hyperactivity disorder).

*Persistence of diagnosis.* Four categories of diagnostic persistence were created based on presence of diagnosis at baseline and/or follow-up: 1) no diagnosis (no diagnosis at both time points), 2) incident (no diagnosis at baseline and presence of diagnosis at follow-up), 3) remittent (presence of diagnosis at baseline and no diagnosis at follow-up), 4) persistent (presence of diagnosis at both time points).

*Impact of mental health problems at follow-up.* was measured according to the 'impact supplement' of the Strength and Difficulties Questionnaire (SDQ) which is part of DAWBA. This supplement assesses the impact of behavioural and emotional difficulties on adolescent's lives

according to guardian reports. A total score (0–10) was generated by summing 5 items: distress, social impairment in family life, friendships, learning, and leisure activities [35]. Higher scores represent greater impact. The impact score has demonstrated internal consistency, cross-informant correlations, and stability measured across time [35].

**Parent-reported stigma towards mental health problems.** To assess parental stigma, we applied the Brazilian Portuguese version of the Reported and Intended Behaviour Scale (RIBS-BP) [36, 37]. The intended behaviour subscale assesses future intended stigmatising behaviour across four domains: living with, working with, living nearby and continuing a relationship with someone with a mental health problem. Higher scores represent lower stigma. The RIBS-BP has demonstrated good internal consistency, and good to excellent construct validity [37].

**Service use.** The Service Assessment for Children and Adolescents (SACA) [38] was used to ask guardians about service contacts made in the past 12 months in response to concerns regarding their child's emotions and behaviour, including alcohol and drugs. The SACA assesses type, nature, frequency and duration of services used, treatments received and settings in which services were delivered. Overall concordance between parent report and records (kappa = 0.76) [38] and test-retest reliability for 12-month (kappa = 0.75–0.86) service use on the parent version of the SACA is strong [39].

We received permission from the SACA developers to translate and adapt the instrument to the Brazilian context in consultation with experts in the Brazilian mental health system to ensure we covered the relevant service types and settings in Brazil. The list of services and professionals was grouped into three sectors: 1) health care: inpatient services (psychiatric hospital, psychiatric unit in a general hospital, alcohol and drug clinic); outpatient services (Centre for psychosocial care [CAPS], which are the community mental health services in Brazil; mental health clinics; specialist mental health professionals (psychiatrists and psychologists in settings other than CAPS and mental health clinics); general health services and professionals (emergency room, paediatrician, general practitioner [GP] or family doctor); 2) education: special school and special education in regular school (special room and special needs class assistant); 3) social care and criminal justice: overnight stay in a shelter or detention centre; probation programme contact; and home visit of the guardianship council (services responsible for child-rights protection).

**Estimation of costs.** Data collected on use of services from the BHRC were combined with unit costs to derive service use costs in Brazilian Reals for the financial year 2018 and then converted to US dollars (based on December 31 2018 conversion rate 1 Real = 0.2581 dollars, according to the Brazilian Central Bank) [40].

*Unit costs*. Detailed information on source of information and unit cost values for each service is available in S1 Table. Where available, we applied unit costs previously reported in the Brazilian literature [41, 42]. However, as costs of many services have not previously been reported, we performed a thorough consultation process gathering relevant data from public databases of the Ministries of Education and Health, and the social care departments of the municipalities of Porto Alegre and São Paulo (S1 Table).

Unit costs were attached to data on service use frequencies for each type of service (based on the SACA) based on 2018 prices or the latest available year converted to 2018 prices using the Nationwide Consumer Price Index. The Brazil Central Bank's calculator was used to apply the index [43]. Once obtained, information on the unit cost of each service was used to calculate the total annual cost by sector (health, education, social care and criminal justice) for each participant by multiplying the frequency of use (e.g. number of visits, nights) by unit cost.

## Data analysis

Data were analysed using STATA, version 14. First, we described prevalence of socio-demographic and clinical characteristics overall and by persistence of psychopathology. Between-group differences were compared using chi-squared tests. For interval variables, means and standard deviations were calculated and overall significance was tested using one-way analysis of variance. A significance parameter of $p < .05$ (two-tailed) was applied for all tests.

Unadjusted odds ratios and coefficients for each predictor and covariate in relation to mental health service use and costs are presented in S2 and S3 Tables. To compare the relative impact between our three main predictors (i.e., psychopathological trajectories, impact of the disorder and parental stigma) of service use and costs we also present logistic regression models for each of these variables adjusting for sociodemographic characteristics (gender, age, mother's education, ethnicity and SEG) and dummy variables (mode of data collection and city of residence) (S4–S6 Tables for service use and S7–S9 Tables for costs). We then used multivariable analyses to examine the association between guardian and adolescent characteristics with service use (logistic regression models) and associated costs (generalised linear models–GLM), overall and by sector: 1) health; 2) education; and 3) social care and criminal justice. All multivariable analyses were adjusted by socio-demographic characteristics, mode of data collection and city. For costs GLM, we analysed the subset of participants who used services in the previous 12 months (n = 143). Annual costs for each participant were included in the models as a scalar dependent variable, with a Gamma distribution [44], using the log-link function.

## Results

Table 1 describes sociodemographic and clinical characteristics of participants. The sample comprised 1,400 adolescents with a mean age of 14.51 years (s.d = 1.98). The majority were white males from low SEG, and only 10.6% of mothers had university education. 23.3% (n = 326) of adolescents had a psychiatric disorder in the previous 12 months, of which 177 (54.3%) were incident and 149 (45.7%) persistent cases since baseline. 213 (15.2%) participants had remitted from a baseline psychiatric diagnosis. 73 (22.4%) of those who presented with a psychiatric disorder (32 incident and 41 persistent cases) reported using some type of service for their mental health in the previous twelve months. The proportion of service use among those who presented a persistent psychiatric condition was 27.5%. Unadjusted odds ratios of any service use among participants with persistent diagnosis were 7.22 (95%CI = 4.50–11.58, p<0.001) compared with participants with no diagnosis, OR = 1.72 (95%CI = 1.02–2.91, p = 0.043) compared with incident and OR = 2.62 (95%CI = 1.52–4.49, p<0.001) with remittent diagnosis. Table 1 present mean disorder impact by psychiatric trajectories. Unadjusted generalised regression models showed that persistent cases presented greater mean difference in disorder impact (SDQ scores) by 2.34, (95%CI = 2.11–2.58, p<0.001) compared with no diagnosis, 1.14 (95% CI = 0.85–1.42, p<0.001), compared with incident cases, and 1.84 (95% CI = 1.56–2.12, p<0.001), compared with remittent cases. Table 1 also describes the mean costs of mental health-related service use in the past year, by psychiatric trajectory (from no diagnosis to persistent psychiatric diagnosis). Bivariate analyses showed a non-significant association between psychiatric trajectory and mean annual costs.

### Frequency of mental health-related service use and annual service use costs

Utilisation of mental health services in the previous 12 months and associated cost by type of service are presented in Table 2. Overall, 10.2% of the sample (n = 143) used some sort of health, education, criminal justice or social care service for mental health problems. Disaggregating by sectors, the health sector had highest proportion of service users (9%), while the

**Table 1. Sociodemographic and clinical characteristics by trajectories of psychopathology (n = 1,400).**

| | No psychiatric diagnosis (n = 861) | Incident psychiatric diagnosis (n = 177) | Remittent psychiatric diagnosis (n = 213) | Persistent psychiatric diagnosis (n = 149) | Overall sample (n = 1,400) | |
|---|---|---|---|---|---|---|
| | N (%) | N (%) | N (%) | N (%) | N (%) | p |
| *Sociodemographic characteristics* | | | | | | |
| Male gender | 503 (58.4) | 81 (45.8) | 134 (62.9) | 83 (55.7) | 801 (57.2) | **0.005** |
| Female gender | 358 (41.6) | 96 (54.2) | 79 (37.1) | 66 (44.3) | 599 (42.8) | |
| Age, mean (s.d) | 14.50 (2.02) | 14.58 (1.90) | 14.39 (1.88) | 14.67 (1.99) | 14.51 (1.98) | 0.564 |
| High SEG | 359 (41.7) | 63 (35.6) | 71 (33.3) | 61 (40.9) | 554 (39.6) | 0.095 |
| Low SEG | 502 (58.3) | 114 (64.4) | 142 (66.7) | 88 (59.1) | 846 (60.4) | |
| White ethnicity | 484 (56.2) | 106 (60.2) | 116 (54.5) | 84 (57.1) | 790 (56.6) | 0.704 |
| Non-White ethnicity | 377 (43.8) | 70 (39.8) | 97 (45.5) | 63 (42.9) | 607 (43.5) | |
| *Guardians characteristics* | | | | | | |
| Maternal no/basic education | 387 (45.1) | 78 (44.6) | 96 (45.3) | 59 (39.9) | 620 (44.5) | 0.953 |
| Maternal secondary education | 384 (44.8) | 78 (44.6) | 93 (43.9) | 71 (48.0) | 626 (44.9) | |
| Maternal university education | 87 (10.14) | 19 (10.9) | 23 (10.9) | 18 (12.2) | 147 (10.6) | |
| *Clinical characteristics* | | | | | | |
| Any Psychiatric Diagnosis | - | 177 (54.3) | - | 149 (45.7) | 326 (23.3) | **<0.001** |
| Fear-related | - | 92 (52.0) | - | 72 (48.3) | 164 (11.7) | **<0.001** |
| Distress-related | - | 70 (40.0) | - | 60 (40.3) | 130 (9.3) | **<0.001** |
| Externalising | - | 49 (27.7) | - | 68 (45.6) | 117 (8.4) | **0.001** |
| SDQ impact mean score (s.d) | 0.28 (0.73) | 1.49 (1.91) | 0.78 (1.51) | 2.62 (2.41) | 0.78 (1.52) | **<0.001** |
| *Mental health-related service use* | | | | | | |
| 12-months service use | 43 (5.0) | 32 (18.0) | 27 (12.7) | 41 (27.5) | 143 (10.2) | **<0.001** |
| Mean service use costs USD$ (s.d) | 326.41 (395.53) | 581.90 (1360.19) | 644.35 (795.50) | 628.50 (901.02) | 527.14 (908.10) | 0.400 |

Notes: Results in bold are significant. SEG, socioeconomic group; SDQ, Strength and Difficulties Questionnaire. 3 missing data in ethnicity variable, 10 missing data in maternal education variable.

education and social care and criminal justice sectors were less frequently contacted with a 1.8% and 1.3% of users respectively. Within the heath sector, the outpatient mental health services, most notably psychologists and psychiatrists in settings other than community mental health clinics, were the most frequently used services/professionals. Inpatient services and general health services such as GP/family doctor, paediatrician and emergency department, were less frequently used. In the education sector, school assistant was the most type of service used by young people, while guardianship council was the most frequently social care service contacted. The total cost of 12-month mental health-related service use for the public purse was 70,110.23 USD. The sector that presented higher total annual cost was the health sector, followed by the education and finally the social care and criminal justice sectors. The services that generated the greatest total costs for the heath sector were psychologist (11,339.64 USD) and CAPS (9,628.01 USD). Among those who used services, the average annual cost of service use amounted to 527.14 USD (SD = 908.10 USD, range = 8.77–7,605.58 USD, median = 221.10 USD, interquartile range = 545.28) per user. Individuals using CAPS (specialty mental health) services (1.1% of the sample) had the highest mean number of visits per person during the

**Table 2. 12-month mental health-related service use and costs by type of service (n = 143).**

| Type of service | Users n (%) | Number of visits/ nights Total | Number of nights/visits per user[a] Mean (Range; s.d.) | Total annual cost per service USD[a,b] | Annual cost per user Mean (Range; s.d.) |
|---|---|---|---|---|---|
| *Health Sector* | | | | | |
| *Inpatient mental health services[3]* | | | | | |
| Psychiatric hospital | 7 (0.5) | 73 | 10.6 (1–30;11.87) | 4,015.72 | 573.67 (66.42–1,992.72;691.76) |
| Psychiatric unit in general hospital | 1 (0.1) | 1 | 1 (1) | 40.49 | 40.49 |
| AD clinic | 3 (0.2) | 48 | 16 (6–27;10.73) | 1,767.90 | 589.30 (191.87–1,096.36;462.10) |
| *Outpatient mental health services* | | | | | |
| Centre for psychosocial care (CAPS) | 15 (1.1) | 452 | 30.15(1–180;49.38) | 9,628.01 | 740.62 (24.56–4,421.03;1212.73) |
| Mental Health clinic | 17 (1.2) | 308 | 18.13(1–70;19.08) | 5,644.64 | 352.79 (19.46–1,362.50;371.28) |
| Psychiatrist | 33 (2.4) | 217 | 6.56 (1–48;8.95) | 5,803.90 | 181.37(27.64–1,326.60;247.29) |
| Psychologist | 71 (5.1) | 1,081 | 15.23 (1–60;14.97) | 11,339.64 | 171.81(11.28–676.99;168.91) |
| AD clinic | 2 (0.1) | 2 | 1 (1) | 14.74 | 14.74 |
| *General Health* | | | | | |
| Emergency department | 4 (0.3) | 9 | 2.25 (1–4;1.50) | 156.0 | 39.00 (17.34–69.34;26.00) |
| Paediatrician | 3 (0.2) | 10 | 3.33 (2–4;1.16) | 120.54 | 40.18 (24.11–48.22;13.92) |
| GP/family doctor | 5 (0.4) | 23 | 4.60 (2–9;2.97) | 403.25 | 80.65 (35.07–157.80;52.01) |
| *Overall health service use* | 126 (9.0) | | | 37,679.94 | 324.83 (11.28–4575.70;590.55) |
| Educational sector[4] | | | | | |
| Special School | 7(0.5) | | School Year | 8,564.92 | 1,223.56 (1,155.72–1,250.70; 44.53) |
| Special Class | 5 (0.4) | | School Year | 6,063.55 | 1,212.71 (1,155.72–1,250.70; 52.02) |
| School Assistant | 12 (0.9) | | School Year | 14,723.52 | 1,226.96 (1,155.72–1,250.70; 42.95) |
| *Overall education service use* | 23 (1.8) | | | 29,351.94 | 1,276.17 (1,155.72–2,501.40; 270.73) |
| **Social care and criminal justice sector** | | | | | |
| Shelter | 2 (0.1) | 210 | 105 (90–120;21.21) | 5,599.95 | 2,799.98 (2,755.34–2,888.48; 63,12) |
| Guardianship Council home visit | 11 (0.8) | 31 | 2.85 (1–5;1.73) | 201.84 | 25.23 (8.77–43.87;15.15) |
| Probation programme | 8 (0.6) | | Six months | 1,875.48 | 234.44 |
| *Overall social care and criminal justice related service use* | 18 (1.3) | | | 4,687.47 | 334.82 (8.77–2,888.48;1,155.72) |

[a]Total cost health sector N = 116, Total cost education sector, N = 23, Total cost social care and criminal justice sector, N = 14. Total cost, N = 133. Cases with missing values in 'frequency of visits' were not included in costs estimates: CAPS = 2, mental health clinic = 1, psychiatrist = 1, psychologist = 5, AD clinic = 1, guardianship council = 3.

[b]Costs are expressed U.S. Dollars, 2018 prices. Brazilian Central Bank conversion rate: Brazilian Real = 0.2581, December 31st 2018[40]

previous year and the highest associated costs among health services. The second highest mean costs in the health sector were related to hospitalizations in psychiatric hospitals and alcohol and drugs clinics, while the lowest mean costs were attributed to emergency department, paediatrician, outpatient alcohol and drugs and GP/family doctor contacts. Although only 0.1% of individuals used shelters, this type of social service had the highest associated mean cost. Education services were used by 1.8% of individuals and these services had the second highest associated mean costs.

## Characteristics associated with mental health-related service use

Having an incident, remittent or persistent psychiatric disorder, as well as the higher impact of behavioural and emotional difficulties on the adolescents' lives and lower parental stigma, all predicted higher odds of any 12-month service use in unadjusted analyses (S2 Table), in models adjusted by sociodemographic characteristics (S4–S6 Tables) and multivariable analyses (Table 3). Service contacts in the health sector were also predicted by the same factors. Service use in the educational sector was predicted by diagnosis trajectory, impact and lower stigma in the unadjusted analyses (S2 Table), and in the models adjusted by sociodemographic characteristics (S4–S6 Tables). However, disorder persistence did not remain significant in multivariable analyses, where impact, lower parental stigma and low SEG showed a significant association with education service use (Table 3). Although persistence of the disorder and higher impact of behavioural and emotional difficulties on the adolescents' lives were associated with social care and criminal justice service use in the unadjusted analyses (S2 Table), and in the models adjusted by sociodemographic characteristics (S4 and S5 Table), there were no factors significantly associated with use of social care and criminal justice services in multivariable analyses (Table 3).

## Characteristics associated with greater mental health related service use costs

When all three sectors were combined into a single total cost variable, greater impact and lower parental stigma were associated with higher costs in unadjusted analyses (S3 Table), in

**Table 3. Multivariable logistic regression models: Predictors of 12-month mental health service utilisation (n = 1,390[a]).**

| Predictors | Any service use | | Health service use | | Education service use | | Social care and criminal justice service use | |
|---|---|---|---|---|---|---|---|---|
| | AOR (95%CI) | p | AOR (95%CI) | p | AOR (95%CI) | p | AOR (95%CI) | p |
| *Sociodemographic characteristics* | | | | | | | | |
| Male gender (Ref) | - | | - | | - | | - | |
| Female gender | 0.87 (0.59–1.28) | 0.488 | 0.98 (0.65–1.46) | 0.901 | 0.67 (0.24–.84) | 0.435 | 1.28 (0.48–3.40) | 0.627 |
| Age (in years) | 1.02 (0.92–1.12) | 0.703 | 1.00 (0.90–1.11) | 0.942 | 0.98 (0.78–1.25) | 0.886 | 1.25 (0.97–1.60) | 0.084 |
| High SEG (Ref) | | | | | | | | |
| Low SEG | 1.30 (0.86–1.98) | 0.211 | 1.11 (0.72–1.70) | 0.646 | **4.31 (1.29–14.39)** | **0.018** | 2.97 (0.75–11.77) | 0.122 |
| White ethnicity (Ref) | - | | - | | - | | - | |
| Non-White ethnicity | 1.17 (0.79–1.72) | 0.442 | 0.97 (0.64–1.46) | 0.869 | 0.53 (0.19–1.48) | 0.225 | 2.59 (0.92–7.28) | 0.071 |
| *Guardians characteristics* | | | | | | | | |
| Maternal no/basic education (Ref) | - | | - | | - | | - | |
| Maternal secondary education | 1.23 (0.82–1.85) | 0.315 | 1.30 (0.84–1.99) | 0.238 | 1.77 (0.66–4.78) | 0.257 | 0.51 (0.17–1.54) | 0.233 |
| Maternal university education | 1.14 (0.59–2.20) | 0.698 | 1.17 (0.59–2.31) | 0.658 | 1.33 (0.24–7.53) | 0.744 | 1.02 (0.19–5.54) | 0.981 |
| Lower parental stigma (RIBS scores) | **1.12 (1.05–1.20)** | **0.001** | **1.11 (1.03–1.18)** | **0.003** | **1.22 (1.01–1.48)** | **0.042** | 1.01 (0.94–1.25) | 0.251 |
| *Clinical characteristics* | | | | | | | | |
| No psychiatric diagnosis (Ref) | - | | - | | - | | - | |
| Incident psychiatric diagnosis | **2.49 (1.44–4.30)** | **0.001** | **2.57 (1.45–4.58)** | **0.001** | 2.29 (0.51–10.97) | 0.281 | 2.54 (0.61–10.52) | 0.199 |
| Remittent psychiatric diagnosis | **2.16 (1.27–3.69)** | **0.005** | **2.22 (1.25–3.93)** | **0.006** | 3.24 (0.84–12.50) | 0.087 | 1.98 (0.45–8.75) | 0.369 |
| Persistent psychiatric diagnosis | **3.01 (1.69–5.36)** | **<0.001** | **3.33 (1.82–6.08)** | **<0.001** | 2.82 (0.65–12.37) | 0.168 | 3.65 (0.88–15.09) | 0.073 |
| SDQ impact score | **1.32 (1.19–1.47)** | **<0.001** | **1.32 (1.19–1.47)** | **<0.001** | **1.51 (1.24–1.84)** | **<0.001** | 1.22 (0.97–1.55) | 0.096 |
| Test statistics | LR x$^2$(13) = 129.35 p<0.001 | | LR x$^2$(13) = 122.81, p<0.001 | | LR x$^2$(13) = 57.46, p<0.001 | | LR x$^2$(13) = 28.36, p = 0.008 | |
| | Pseudo-R$^2$ = 0.14 | | Pseudo-R$^2$ = 0.15 | | Pseudo-R$^2$ = 0.25 | | Pseudo-R$^2$ = 0.15 | |

[a]From the total sample, N = 1,400, 10 cases had missing data in mother's education and 3 in ethnicity variables. AOR = Adjusted odds ratios. Results in bold are statistically significant (p<0.05). Models adjusted by collection instrument and city.

the models adjusted by sociodemographic characteristics (S8 and S9 Tables) and multivariable analyses (Table 4). White ethnicity was also associated with higher costs in multivariable analyses (Table 4). Each additional impact score predicted an increase in mean costs of 142.59 USD (p<0.001). For parental stigma, each additional RIBS-BP score (indicating lower stigma) increased mean cost by 69.32 USD (p = 0.020). White ethnicity was associated with having higher mean costs of 295.49 USD (p = 0.036), compared with non-white participants. No association was found between broad diagnosis categories and costs (S10 Table).

When looking at predictors of costs according to sector (Table 4), disorder impact was associated with greater health sector service use (predicted mean cost by each impact score = 66.26 USD, p = 0.019). We did not find any significant association of psychiatric trajectories, impact of disorder or parental stigma with education sector's costs (S3, S7–S9 Tables, and Table 4). Although persistence of the disorder was associated with social care and criminal justice service's costs in the unadjusted analyses (S3 Table), and in the models adjusted by sociodemographic characteristics (S7 Table), we did not find any significant factors associated with social care/criminal justice sectors' costs in multivariable analyses (Table 4).

## Discussion

We analysed data on mental health-related service use and associated costs among a prospective community cohort of young people in Brazil. We found that impact of mental health

**Table 4. Generalised linear models: Parental and clinical characteristics associated with cost of mental health service use in the last 12 months, overall and by sector.**

| Predictors | Any service use N = 131 | | Health service use N = 115 | | Education service use N = 22 | | Social care and criminal justice service use N = 14 | |
|---|---|---|---|---|---|---|---|---|
| | β (95%CI) | p | β (95%CI) | p | β (95%CI) | p | β (95%CI) | p |
| *Sociodemographic characteristics* | | | | | | | | |
| Male gender (Ref) | - | | - | | - | | - | |
| Female gender | 0.05 (-0.50–0.59) | 0.866 | 0.06 (-0.61–0.73) | 0.854 | -0.03 (-0.29–0.24) | 0.857 | 14.41 (-5.34–34.17) | 0.153 |
| Age (in years) | -0.05 (-0.19–0.10) | 0.522 | 0.06 (-0.14–0.24) | 0.572 | 0.02 (-0.02–0.06) | 0.886 | -2.11 (-4.87–0.65) | 0.133 |
| High SEG (Ref) | | | | | | | | |
| Low SEG | 0.47 (-0.08–1.03) | 0.092 | -0.13 (-0.78–0.53) | 0.706 | -0.03 (-0.36–0.29) | 0.839 | 3.32 (-8.52–15.15) | 0.583 |
| White ethnicity (Ref) | - | | - | | - | | - | |
| Non-White ethnicity | **-0.55 (-1.07- -0.04)** | **0.036** | -0.12 (-0.75–0.51) | 0.707 | 0.09 (-0.10–0.27) | 0.368 | -4.28 (-9.80–1.24) | 0.129 |
| *Guardians characteristics* | | | | | | | | |
| Maternal no/basic education (Ref) | - | | - | | - | | - | |
| Maternal secondary education | 0.27 (-0.29–0.82) | 0.341 | -0.10 (-0.78–0.58) | 0.776 | -0.07 (-0.23–0.10) | 0.418 | 4.95 (-2.19–12.08) | 0.174 |
| Maternal university education | 0.003 (-0.90–0.91) | 0.995 | -0.34 (-1.38–0.69) | 0.515 | 0.40 (-0.07–0.87) | 0.094 | - | - |
| Lower parental stigma (RIBS score) | **0.12 (0.12–0.39)** | **0.020** | 0.04 (-0.07–0.16) | 0.465 | 0.002 (-0.06–0.06) | 0.948 | 0.05 (-0.98–1.08) | 0.922 |
| *Clinical characteristics* | | | | | | | | |
| No psychiatric diagnosis (Ref) | - | | - | | - | | - | |
| Incident psychiatric diagnosis | -0.14 (-0.83–0.55) | 0.693 | 0.15 (-0.71–1.00) | 0.735 | 0.07 (-0.15–0.29) | 0.548 | -23.61 (-54.48–7.27) | 0.134 |
| Remittent psychiatric diagnosis | 0.39 (-0.35–1.14) | 0.298 | 0.09 (-0.85–1.04) | 0.847 | -0.01 (-0.23–0.21) | 0.928 | -2.17 (-7.42–3.07) | 0.417 |
| Persistent psychiatric diagnosis | -0.39 (-1.16–0.38) | 0.315 | -0.42 (-1.40–0.58) | 0.412 | 0.14 (-0.11–0.39) | 0.276 | -17.23 (-36.84–2.39) | 0.085 |
| SDQ impact score | **0.25 (0.12–0.39)** | **<0.001** | **0.20 (1.19–1.47)** | **0.019** | 0.01 (-0.02–0.04) | 0.458 | -0.34 (-1.51–0.83) | 0.569 |
| Test statistics[a] | AIC 16.97193 | | AIC 16.24671 | | AIC 20.26365 | | AIC 15.83608 | |
| | BIC -353.9633 | | BIC -308.0671 | | BIC -24.59063 | | BIC 1.230216 | |
| | R² = 0.22 | | R² = 0.15 | | R² = 0.79 | | R² = 0.90 | |

Notes: Results in bold are significant (p<0.05). Models adjusted by city and method of interview.

[a]Cameron & Windmeijer's *R*-squared, measure of goodness of fit for the class of exponential family regression models.

problems on daily life and parental stigma were the most consistent and robust drivers of mental health service use and associated costs.

## Drivers of mental-health service use costs

The association between disorder impact and mental health-related service use and costs that we found has been observed in previous research, providing further support that impact and impairment tend to be the strongest and most robust predictors of mental health service use [13, 35] and costs [14]. Contrary to what we expected, we did not find an association between disorder persistence and costs. Our analyses instead found that impact of the disorder on adolescent's life was the most important clinical predictor and that this was what seemed to drive service costs rather than type or persistence of diagnosis. This finding suggests that although the trajectory of the diagnosis is associated with a greater likelihood of having any contact with mental health-related services, this association did not translate to costs which also reflects the intensity and / or type of care and number of visits per year. Nevertheless, it is important to consider that we have estimated annual costs, and these do not necessarily reflect the cumulative economic costs of persistent cases across childhood and adolescence.

We also found that lower parental stigma was associated with greater service use and higher costs. Our findings suggest that the ways in which parents perceive mental illness in adolescents may significantly influence help-seeking. We are aware of two studies showing that young people's likelihood of service use across health and education settings was greater among caregivers who reported less intended stigmatising behaviours [19]. Another study indicated that low parental stigmatising attitudes toward mental disorders increased recognition of mental health problems in preadolescents (10–12 years) [45].

We only found one study [20] exploring the impact of parental stigmatising attitudes toward mental illness on child service use costs. The cited study, conducted in the UK, with a smaller sample size, did not find an association between parental stigma and young people's service use costs. Other research has shown that parental stigma can impede problem recognition and help-seeking [16, 45]. Higher stigma amongst parents and caregivers may discourage or delay service access for their children [19], which may reduce the short-term public sector direct costs of treatment but be detrimental in the long run. Future research needs to further explore the mechanisms through which parental stigma may be related to service/treatment selection and treatment adherence, in order to explain its impact on treatment costs. Moreover, as lower parental stigma may facilitate earlier service contact, it would be interesting to investigate if lower parental stigma may result in lower costs in the longer term.

Among sociodemographic variables, we found that low SEG predicted higher odds of educational service use. This may be related to the fact that young people living in deprived circumstances are more likely to be affected by developmental problems [46], and, therefore, are more likely to use special education services [11]. Although our study did not identify any differences in service use according to ethnicity, we found white ethnicity was associated with higher service use costs. This may reflect disparities in the type of mental health treatment offered or available to non-white children/adolescents. According to previous studies, non-white children/adolescents are less likely to receive adequate mental health treatment [47], including lower likelihood of psychopharmacological prescriptions [48], compared with white children/adolescents.

## The economic impact of adolescent mental health care by sectors

We found that the health sector was clearly the main sector accessed by youth with mental disorders. Within the health sector, specialty mental health care was used more frequently and

was more costly than primary care. In Brazil, access to CAPS does not require any referral. However, the number of CAPS services are limited, and they are focused on treatment of severe mental disorders [26]. The high costs incurred by the mental health sector for the treatment of psychiatric disorders in CAPS may be because these services provide intensive outpatient treatments (reflected by the highest number of visits we found), which is costly compared with no-specialized services. It is important to highlight that the lack of youth-oriented primary care mental health services in Brazil limits access to treatment. This could explain why we found low frequency of mental health-related contacts with GP/ family doctors. As a result, contact with specialist mental health services only happens when the disorder has significant negative impact on the lives of young people. In this sense, the organisation of a mental health network of care for adolescents, integrating primary care, social care, education, criminal justice and community youth-specialist services, according to the impact of cases, must be considered in Brazil to adequately plan and allocate scarce public budgets [49].

We found that mental-health related educational service use was less prevalent compared with health service use, nevertheless–as previous studies have shown–[11, 14] educational service use was also associated with higher costs. In Brazil, while special education services are provided in regular schools, their use is restricted to students with disabilities and developmental disorders [50], so only adolescents with severe mental disorders are likely to be eligible.

## Limitations

Our study has several limitations. First, the psychometric properties of the adapted version of the SACA have not been evaluated yet. Second, as we were not able to access administrative records, service use assessment was limited to guardians' reports. However, the concordance between parent report and records for service use on the parental version of the SACA is strong [38]. Third, as most of the unit cost were specifically identified for the cities where the BHRC is being conducted, São Paulo and Porto Alegre, they are not necessarily generalisable to the whole country. Fourth, due to the limited number of participants using each type of service, we were unable to compare factors related with use and associated costs of specific types of service. Fifth, due to an administrative error we were unable to contact the first 6% of first follow-up BHRC participants. This could reduce our sample size, but as this was a random error, we do not believe that it affected the results. Furthermore, given our estimates come from observational cohort data, we are not able to establish causality.

## Conclusions

Our findings suggest that the main drivers of health-related service use costs among adolescents in Brazil were impact of mental health problems, in addition to lower stigma toward people with mental illness among guardians and White ethnicity. In the present study, only 22.4% of young people with a diagnosed mental disorder received any form of care. In addition to reducing inequality in service use among children, our findings also argue for lowering barriers to care, in particular addressing caregiver stigma. Furthermore, because lower use of services in adolescence may be associated with worse outcomes across the life course [49], further reducing barriers to service utilisation by young people is important, even though this may imply higher short-term costs.

Guardian's lower stigmatising attitudes towards mental disorders may be crucial to support young people in accessing, engaging and maintaining contact with mental health-related services. Various anti-stigma interventions have demonstrated effectiveness for improving help-seeking [51], but few have been implemented in LMICs. Further studies are needed to design and implement anti-stigma interventions in LMICs. On the other hand, health and education

policies need to better support guardians to access appropriate and timely services in their communities, before the symptoms have a significant impact on adolescent functioning. We conclude that the organisation of a culturally sensitive mental health network of care for adolescents, integrating primary care, social care, education, criminal justice services and CAPS, must be considered in Brazil to adequately plan and allocate scarce public budgets.

## Supporting information

**S1 Fig. Flow chart of Brazilian High-Risk Cohort participants included in the mental-health related service use study.**
(PDF)

**S1 Table. Unit costs of health, educational, social care and criminal justice related services.**
(PDF)

**S2 Table. Bivariate analysis: Predictors of 12-month mental health service utilisation.**
(PDF)

**S3 Table. Bivariate analysis: Predictors of cost of mental health service use in the last 12 months.**
(PDF)

**S4 Table. Logistic regression models: 12-month mental health service utilisation predicted by psychiatric diagnosis trajectories.**
(PDF)

**S5 Table. Logistic regression models: 12-month mental health service utilisation predicted by impact of behavioural and emotional difficulties on child' life.**
(PDF)

**S6 Table. Logistic regression models: 12-month mental health service utilisation predicted by parental stigma.**
(PDF)

**S7 Table. Generalised linear models: Cost of 12-month mental health service utilization predicted by psychiatric diagnosis trajectories.**
(PDF)

**S8 Table. Generalised linear models: Cost of 12-month mental health service utilisation predicted by impact of behavioural and emotional difficulties on child' life.**
(PDF)

**S9 Table. Generalised linear models: Cost of 12-month mental health service utilisation predicted by parental stigma.**
(PDF)

**S10 Table. Generalised linear models: Cost of 12-month mental health service utilisation predicted by broad diagnosis categories.**
(PDF)

## Author Contributions

**Conceptualization:** Carolina Ziebold, Wagner Silva-Ribeiro, Giovanni Abrahão Salum, Sara Evans-Lacko.

**Data curation:** Carolina Ziebold.

**Formal analysis:** Carolina Ziebold, Wagner Silva-Ribeiro.

**Funding acquisition:** Eurípedes Constantino Miguel, Rodrigo Affonseca Bressan, Giovanni Abrahão Salum, Jair de Jesus Mari, Sara Evans-Lacko.

**Investigation:** Carolina Ziebold.

**Methodology:** Carolina Ziebold, Wagner Silva-Ribeiro, Derek King, David McDaid, Mauricio Scopel Hoffmann, Renee Romeo, Pedro Mario Pan, Eurípedes Constantino Miguel, Luis Augusto Rohde, Giovanni Abrahão Salum, Jair de Jesus Mari, Sara Evans-Lacko.

**Project administration:** Pedro Mario Pan, Eurípedes Constantino Miguel, Rodrigo Affonseca Bressan, Giovanni Abrahão Salum.

**Supervision:** Wagner Silva-Ribeiro, David McDaid, Jair de Jesus Mari, Sara Evans-Lacko.

**Writing – original draft:** Carolina Ziebold, Sara Evans-Lacko.

**Writing – review & editing:** Carolina Ziebold, Wagner Silva-Ribeiro, Derek King, David McDaid, Mauricio Scopel Hoffmann, Renee Romeo, Pedro Mario Pan, Eurípedes Constantino Miguel, Rodrigo Affonseca Bressan, Luis Augusto Rohde, Giovanni Abrahão Salum, Jair de Jesus Mari, Sara Evans-Lacko.

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
