## [Decision Letter · Decision Letter 0]

1 Dec 2021

PONE-D-21-10663Utilisation and costs of mental health-related service use among adolescentsPLOS ONE

Dear Dr. Evans-Lacko,

Thank you for submitting your manuscript to PLOS ONE and our sincere apologies for the delay in the reviewing process. After careful consideration, we feel that it has merit but does not fully meet PLOS ONE’s publication criteria as it currently stands. Both reviewers have made some suggestions that can improve your manuscript. Therefore, we invite you to submit a revised version of the manuscript that addresses the points raised during the review process.

We look forward to receiving your revised manuscript.

Kind regards,

Therese van Amelsvoort

Academic Editor

PLOS ONE

Journal Requirements:

2. Thank you for including your ethics statement: "This research was carried out in accordance with the latest version of the Declaration of Helsinki. Child assent and parental informed consent was obtained from the research subjects. All procedures were approved by the Ethics Committee of the Federal University of São Paulo-UNIFESP (Nº 2.879.533 and - CAAE 06457219.9.0000.5505), Hospital de Clínicas de Porto Alegre (CAAE 06457219.9.3001.5327) and the European Research Commission" 

a) Please provide additional details regarding participant consent. In the ethics statement in the Methods and online submission information, please ensure that you have specified what type you obtained (for instance, written or verbal, and if verbal, how it was documented and witnessed). If your study included minors, state whether you obtained consent from parents or guardians. If the need for consent was waived by the ethics committee, please include this information.

"The authors CZ, WR, DK, DMD, MH, RR, GS, PP, EM, JM, and SEL have no conflicts of interest to declare. LR has received grant or research support from, served as a consultant to, and served on the speakers’ bureau of Medice, Novartis/Sandoz and Shire/Takeda in the last three years. The ADHD and Juvenile Bipolar Disorder Outpatient Programs chaired by LR have received unrestricted educational and research support from the following pharmaceutical companies in the last three years: Novartis/Sandoz and Shire/Takeda. LR has received authorship royalties from Oxford Press and ArtMed and travel grants from Shire to take part in the 2018 APA annual meeting and from Novartis to take part of the 2017 AACAP annual meeting. RB: Pfizer, Torrent and Sanofi-Aventis."

We note that you received funding from a commercial source: Novartis and Sandoz

Reviewers' comments:

Reviewer's Responses to Questions

**Comments to the Author**

1. Is the manuscript technically sound, and do the data support the conclusions?

Reviewer #1: Yes

Reviewer #2: Yes

2. Has the statistical analysis been performed appropriately and rigorously? 

Reviewer #1: Yes

Reviewer #2: Yes

3. Have the authors made all data underlying the findings in their manuscript fully available?

Reviewer #1: No

Reviewer #2: Yes

4. Is the manuscript presented in an intelligible fashion and written in standard English?

Reviewer #1: Yes

Reviewer #2: Yes

5. Review Comments to the Author

Reviewer #1: General

Very relevant and interesting study. Well written paper, I found it pleasant to read. I would recommend some minor adjustments:

Abstract

- When only reading the abstract, the distinction between incident, remittent and persistent disorder in the Results section is a bit confusing. For the abstract, I would recommend rewriting this sentence for example: “Higher odds of service use were associated with having a diagnosed mental disorder (either incident, remittent or persistent), higher impact of symptoms etc.”

Introduction

- This study focuses on the economic cost of mental disorders in young people (line 52). Therefore, it should be better introduced why, in addition to (mental) health services, also education, criminal justice and social care services were investigated.

- Line 55: male gender is mostly not associated with higher use of mental health services. Please specify the association between these factors and specific services.

Methods: Data and participants

- I understand that not all information about the Brazilian High-Risk Cohort was included in this paper. I would want to know, however, based on what information the children became part of this high risk cohort. Are they COPMI?

Methods: Measures

- Why only maternal educational level?

- Furthermore, this paragraph forms a clear description of appropriate measures.

Results

- Very clear description and informative tables.

Discussion

- Line 325: “We found that the health sector was clearly the main sector providing mental health care for youth.” That’s quite obvious. I would recommend rewriting this, for example: “We found that the health sector was clearly the main sector accessed by youth with mental disorders.”

- In the present study, only 20% of young people with a diagnosed mental disorder received any form of care. In addition to reducing inequality in service use among children, these data also argue for lowering barriers to care for young people in general. I would recommend stating this in the conclusion as well.

- Line 329: “The lack of youth-oriented primary care mental health programmes”. Is this also the reason why GP’s/family doctors were less frequently visited?

- Line 359-361: this reads like the impact of mental health problems on children’s lives should be increased because it would support help-seeking. Please, rewrite.

- Line 363: effectives should be effective

- In future research, it would be interesting to not only assess parental stigma but also stigma among the adolescents themselves.

Reviewer #2: It's good to see more representative research from LMICs, trying to bridge the existing knowledge gap. This study's most significant plus point is that it looks at service use and service cost from multiple angles, shedding light on demographic, clinical and systemic factors that contribute to service use cost. However, this manuscript does require significant improvement in language and content. Here are my main suggestions:

1) The language of the manuscript can be crisper. Multiple places sentences look disjointed or elongated. The paragraphs are changed too frequently in some places, with each of these paragraphs containing only one or two sentences.

2) Introduction: In line 57, please clarify whether by 'education services' authors mean remedial education services or some other kind of services?

3) Introduction: The lines 55-58 are difficult to follow: authors claim that certain demographic and clinical characteristics are associated with a greater likelihood of using certain services as per existing research. However, it's not clear how this connects with the assertion about young people in the same sentence.

4) Introduction: The importance of studying parental stigma needs to be built better.

5) Introduction: I'm not sure what is meant by 'beyond diagnosis', are authors implying the existing studies cover the cost of diagnosis only or for limited kinds of disorders. Some clarification here would be helpful.

6) Introduction: The way lines 72-73 are written makes it sound like Brazil is a high-income country

7) Introduction: In line 88, it's unclear what characteristics the authors are referring to and whether the following hypothesis is related to a subset of these characteristics?

8) Methods: In line 96, some information on how these children were classified as high risk will be helpful. The authors have said the details are somewhere else, but a brief description here will make it easier for the reader to understand the sample.

9) Methods: In line 99, it was slightly hard to follow study timelines. Was this study carried out after the first follow-up in 2014-2015 or as part of the follow-up?

10) Methods: The authors can use consistent terminology: children or young people. As of now, this has varied from one sentence to another.

11) the '-' in line 102 seems typo.

12) Methods: In lines 127-130, it's unclear why young people were not interviewed at baseline but were included during the 3-year follow-up?

13) Methods: Do authors have any psychometric properties of the adapted version of Service Assessment for Children and Adolescents that can be reported in this publication?

14) Results: In line 224, the authors refer to Table 1. However, without any commentary on the significance of data in this table, the authors jump to a new set of findings. All this makes it slightly hard to follow what is being presented.

15) The 12-month service use and service use cost means are presented in Tables 1 and 2. Repeating the same findings across two tables should be avoided

16) The paragraph on page 12 lacks a description of the cost associated with each service? For e.g., although CAPS is not a highly prevalent service, the associated cost makes for a lion contribution to the public purse. This needs to be presented and discussed.

17) Discussion: In line 288, the use of the terms 'above and beyond' doesn't convey much. To the best of my knowledge, the current analysis nowhere helps to reach this conclusion of above and beyond. I am requesting authors to look at terminology closely.

18) Discussion: Lines 301-312 can be streamlined and better organised.

19) Discussion: Line 327: The number of CAPS users was less, but the number of visits and costs for those who used it were very high. These were not reflected in the discussion, nor were its implication for the restructuring health system.

20) Discussion: The hypothesis stated that researchers were interested in examining the impact of persistence of psychiatric disorders from childhood to adolescence on service costs; however, the discussion did not give much attention to this part.

21) Conclusion: Some of the text in the last paragraph of the conclusion, i.e. those referring to implications, can be moved to discussion and expanded further.

I'm not able to comment on cost analysis as this is not my area of expertise.

6. PLOS authors have the option to publish the peer review history of their article (what does this mean?). If published, this will include your full peer review and any attached files.

Reviewer #1: No

Reviewer #2: **Yes: **KANIKA MALIK

---

## [Author Response · Author response to Decision Letter 0]

15 Feb 2022

We appreciate the careful revision of our manuscript and the comments of the reviewers. We are pleased to be invited to submit the revised version of our paper to PLOS ONE. 

Please find attached both an unmarked version of the revised manuscript and one version with changes marked in red. Our point-by-point responses to the reviewers’ comments (unquoted italics) and details of the changes we have performed to our revised manuscript are given below. 

Reviewer #1: 

General Comment: Very relevant and interesting study. Well written paper, I found it pleasant to read. I would recommend some minor adjustments

Response: We appreciate your positive feedback, the careful revision of our manuscript and your comments.

Comment 1: Abstract- When only reading the abstract, the distinction between incident, remittent and persistent disorder in the Results section is a bit confusing. For the abstract, I would recommend rewriting this sentence for example: “Higher odds of service use were associated with having a diagnosed mental disorder (either incident, remittent or persistent), higher impact of symptoms etc.”

Response: Thank you for your comment. We have rewritten this sentence as follows:

Higher odds of service use were associated with having a diagnosed mental disorder (either incident [OR=2.49, 95%CI=1.44-4.30, p=0.001], remittent [OR=2.16, 95%CI=1.27-3.69, p=0.005] or persistent [OR=3.01, 95%CI=1.69-5.36, p<0.001]), higher impact of symptoms.. 

Comment 2: Introduction- This study focuses on the economic cost of mental disorders in young people (line 52). Therefore, it should be better introduced why, in addition to (mental) health services, also education, criminal justice and social care services were investigated.

Response: We have edited the introduction as follows:

The high prevalence and potentially enduring nature of these impacts make addressing youth mental health conditions particularly important, but this is a challenge for public systems with limited resources (Knapp M; Evans-Lacko S, 2015). Economic costs associated with youth mental health conditions involve a wide range of sectors including health, educational, social care, and criminal justice services [9,10]. This can represent a substantial cost to the public purse, yet it could also be considered a wise investment given the evidence that effective treatment can mitigate the impact of poor mental health (Knapp et al., 2011).

Comment 3: Line 55: male gender is mostly not associated with higher use of mental health services. Please specify the association between these factors and specific services.

Response: We appreciate your suggestion. We have edited this paragraph in the revised version of the manuscript:

Some studies from high–income countries suggest that lower socioeconomic status, as well as clinical features (illness severity and impact of disorders) are associated with use of health, special education, and social care services, while male gender and older age are associated with more criminal justice services contacts [11,13,14]. These sociodemographic and clinical characteristics are also associated with greater mental health-related treatment costs among young people [11,14,15]

Comment 4: Methods: Data and participants. I understand that not all information about the Brazilian High-Risk Cohort was included in this paper. I would want to know, however, based on what information the children became part of this high risk cohort. Are they COPMI?

Response: Thanks for the important point you raised. We have added information in the methods on the Brazilian High-Risk cohort sampling procedures as follows: 

This study is nested within the Brazilian High-Risk Cohort (BHRC), which is an ongoing prospective longitudinal study that comprises a community sample and a high‐risk sub‐sample (a sample at increased risk of mental disorders) of young people from Sao Paulo and Porto Alegre, Brazil. A detailed description of the sample and procedures can be found elsewhere [25]. Briefly, during the registry day, 12,500 parents of young people aged 6 to 14 years attending 57 schools (22 in Porto Alegre and 35 in São Paulo) were invited to a screening of mental health disorders using the Family History Screen (FHS) [26]. A total of 8,012 families (9,937 eligible children, 45,394 family members) were interviewed. Based on the percentage of members in the family that screened positively for psychiatric disorders, an index of family load for each potential eligible child was computed. The final cohort comprised 2,511 young people; 957 were randomly selected, and 1,554 were a sub‐sample at increased risk of mental disorders based on the FHS. 

Comment 5: Methods: Measures. Why only maternal educational level?

- Furthermore, this paragraph forms a clear description of appropriate measures.

Response: As stated in the methods section, the socioeconomic group variable comprised head of household educational level in addition to other household socioeconomic indicators. As some research suggests that mothers educational level is particularly important for recognition and help-seeking, we also included this variable as a separate indicator. As the vast majority of caregiver respondents were mothers (in 93% of cases the biological mother [information included in the revised manuscript]) we focused on maternal education rather than estimating the educational level of other caregivers. 

Comment 6: Results. Very clear description and informative tables.

Response: Thank you very much for your positive feedback.

Comment 7: Discussion- Line 325: “We found that the health sector was clearly the main sector providing mental health care for youth.” That’s quite obvious. I would recommend rewriting this, for example: “We found that the health sector was clearly the main sector accessed by youth with mental disorders.”

Response: Thank you very much for your suggestion. We rewrote this sentence as follows: 

We found that the health sector was clearly the main sector accessed by youth with mental disorders.

Comment 8: In the present study, only 20% of young people with a diagnosed mental disorder received any form of care. In addition to reducing inequality in service use among children, these data also argue for lowering barriers to care for young people in general. I would recommend stating this in the conclusion as well.

Response: Thanks for your suggestion. We have edited the first paragraph of the conclusions as follows:

Our findings suggest that the main drivers of health-related service use costs among adolescents in Brazil were impact of mental health problems, in addition to lower stigma toward people with mental illness among guardians and White ethnicity. In the present study, only 22.4% of young people with a diagnosed mental disorder received any form of care. In addition to reducing inequality in service use among children, our findings also argue for lowering barriers to care, in particular addressing caregiver stigma. Furthermore, because lower use of services in adolescence may be associated with worse outcomes across the life course [47], it is needed to further explore measures to reduce inequalities in service utilisation by young people, even though this implies higher short-term costs. 

Comment 9: Line 329: “The lack of youth-oriented primary care mental health programmes”. Is this also the reason why GP’s/family doctors were less frequently visited?

Response: We appreciate your comment, and we agree with your interpretation of this result. We have edited the referred sentence:

The lack of youth-oriented primary care mental health programmes limits access to treatment when symptoms start to have an impact on adolescent functioning. This can explain why we found a low rate of mental health-related contacts with GP/ family doctors. As a result, contact with specialist mental health services only happens when the disorder has significant negative impact on the lives of young people. 

Comment 10: Line 359-361: this reads like the impact of mental health problems on children’s lives should be increased because it would support help-seeking. Please, rewrite.

Response: We have rewritten this paragraph:

Guardian’s lower stigmatising attitudes towards mental disorders may be crucial to support young people in accessing, engaging and maintaining contact with mental health-related services. Various anti-stigma interventions have demonstrated effectiveness for improving help-seeking [49], but few have been implemented in LMICs. Further studies are needed to design and implement anti-stigma interventions in LMICs. On the other hand, health and education policies need to better support guardians to access appropriate and timely services in their communities, before the symptoms have a significant impact on adolescent functioning.

Comment 11: Line 363: effectives should be effective

Response: Thank you very much, we have corrected this error.

Comment 12: In future research, it would be interesting to not only assess parental stigma but also stigma among the adolescents themselves.

Response: We agree with you, and we are planning to evaluate the association between mental health-related service use and youth stigma towards mental illness in future cohort’s assessments.

Reviewer #2: 

General comment: It's good to see more representative research from LMICs, trying to bridge the existing knowledge gap. This study's most significant plus point is that it looks at service use and service cost from multiple angles, shedding light on demographic, clinical and systemic factors that contribute to service use cost. However, this manuscript does require significant improvement in language and content. Here are my main suggestions:

Response: We appreciate your positive opinion of our work, the careful revision of our manuscript and your valuable comments.

Comment 1: The language of the manuscript can be crisper. Multiple places sentences look disjointed or elongated. The paragraphs are changed too frequently in some places, with each of these paragraphs containing only one or two sentences.

Response: Thanks for your comment. We have revised and edited the language through the manuscript. 

Comment 2: Introduction: In line 57, please clarify whether by 'education services' authors mean remedial education services or some other kind of services?

Response: Thanks for your comment. We have indicated ‘special education’ in the revised version of the manuscript.

Comment 3: Introduction: The lines 55-58 are difficult to follow: authors claim that certain demographic and clinical characteristics are associated with a greater likelihood of using certain services as per existing research. However, it's not clear how this connects with the assertion about young people in the same sentence.

Response: Thanks for your comment. We have edited and separated these sentences:

Some studies from high–income countries suggest that lower socioeconomic status, as well as clinical features (illness severity and impact of disorders) are associated with use of health, special education, and social care services, while male gender and older age are associated with more criminal justice service contacts [11,13,14]. These sociodemographic and clinical characteristics are also associated with greater mental health-related treatment costs among young people [11,14,15].

Comment 4: Introduction: The importance of studying parental stigma needs to be built better.

Response: We appreciate your suggestion. We have included the following changes:

Families also play a central role in young people’s contact with services. One study from the UK found that lower mental illness-related stigma among caregivers was associated with an increased likelihood of young people’s mental health service use [16]. Stigmatising attitudes toward mental illness amongst parents may influence service contacts due to shame and fears of labelling their child’s mental health condition [16]. There are clear links between stigma and reduced help-seeking [17], reduced adherence to treatment and early withdrawal from services [17,18]. However, little is known about/ how parental stigma could impact on young people service use and costs.

Comment 5: Introduction: I'm not sure what is meant by 'beyond diagnosis', are authors implying the existing studies cover the cost of diagnosis only or for limited kinds of disorders. Some clarification here would be helpful.

Response: Thanks for your suggestion. We have edited this sentence as follows:

Additionally, little is known about how, in addition to the type of disorder, whether persistence of psychopathology from childhood to adolescence, disorders’ impact on adolescent’s daily life (i.e., functioning), and key barriers to care such as stigma, could influence costs. 

Comment 6: Introduction: The way lines 72-73 are written makes it sound like Brazil is a high-income country

Response: We appreciate your comment. We have deleted ‘Similar to most high income countries’ in the revised version of the manuscript. 

Comment 7: Introduction: In line 88, it's unclear what characteristics the authors are referring to and whether the following hypothesis is related to a subset of these characteristics?

Response: We have rewritten this sentence to clarify the characteristics under study:

Second, we examine how costs vary according to: mental health trajectories, impact of the disorder on everyday life, and parent/guardian stigma towards mental illness.

Comment 8: Methods: In line 96, some information on how these children were classified as high risk will be helpful. The authors have said the details are somewhere else, but a brief description here will make it easier for the reader to understand the sample.

Response: Thanks for your suggestion. As explained in response to Reviewer 1’s comment 4, we have included a brief description of the Brazilian High-Risk Cohort sampling procedures.

Comment 9: Methods: In line 99, it was slightly hard to follow study timelines. Was this study carried out after the first follow-up in 2014-2015 or as part of the follow-up?

Response: We have tried to clarify this including the following information:

Cohort participants were interviewed at baseline (aged 6-14 years, calendar year:2010-2011, n=2,511), and at first follow-up (N=2010, aged 9-17 years, calendar year 2014). After completing the BHRC first follow-up interview, 1,881 parents/guardians were invited to respond to a supplementary interview which included a comprehensive assessment of mental health related service use (calendar year: 2014-2015, young people participants aged 10-18 years).

Comment 10: Methods: The authors can use consistent terminology: children or young people. As of now, this has varied from one sentence to another.

Response: Thanks for your comment. We have revised and edited the methods section in order to use consistently the term young people. 

Comment 11: The '-' in line 102 seems typo.

Response: We appreciate your comment. We have deleted this typo. 

Comment 12: Methods: In lines 127-130, it's unclear why young people were not interviewed at baseline but were included during the 3-year follow-up?

Response: This was because participants were younger at baseline and so we relied on parent’s report, given limitations in funding and resources. Given that older adolescents are better at reporting internalising symptoms, both guardian and youth interviews were performed at 3-year follow-up. We included this explanation in the revised version of the manuscript:

At baseline, diagnostic assessment and interviews were performed with guardians only. Previous literature has found that self-reports on internalising conditions during adolescence is higher compared with parental report. This can be explained because internalising problems, such as anxiety or depression, would be less observable by guardians, being advisable to consider both reports to reach a reliable evaluation of adolescent mental health [30,31]. For this reason, diagnostic assessment at 3-year follow-up was performed considering guardian reports and additional information from interviews with the young people about internalising conditions.

Comment 13: Methods: Do authors have any psychometric properties of the adapted version of Service Assessment for Children and Adolescents that can be reported in this publication?

Response: 

The parent-report SACA has been shown to be a valid measure of young people’s service use (kappa = 0.76; [Hoagwood et al., 2000]) with test-retest reliability for past-year reports (ranging from 0.75 to 0.86; [Horwitz et al., 2001]). We have not assessed the psychometric properties of the adapted version of the Service Assessment for Children and Adolescents for Brazilian participants yet. We have included this limitation in the revised version of the manuscript. 

Comment 14: Results: In line 224, the authors refer to Table 1. However, without any commentary on the significance of data in this table, the authors jump to a new set of findings. All this makes it slightly hard to follow what is being presented.

Response: We appreciated your comment. We have edited this paragraph: 

Table 1 describes sociodemographic and clinical characteristics of participants. The sample comprised 1,400 adolescents with a mean age of 14 years (s.d=1.98). The majority were white males from low SEG, and only 10% of mothers had university education. 23.3% (n= 326) of adolescents had a psychiatric disorder in the previous 12 months, of which 177 (54.3%) were incident and 149 (45.7%) persistent cases since baseline. 213 (15.2%) participants had remitted from a baseline psychiatric diagnosis. Participants with externalising disorders were more likely to have persistent trajectories (RR=2.19, 95%CI=1.38-3.48, p<0.001). Participants categorised as persistent also reported greater disorder impact (�=2.34, 95%CI=2.11-2.58, p<0.001). 22.4% of those who presented with a psychiatric disorder reported using some type of service for their mental health in the previous twelve months. The proportion of service use among those who presented a persistent psychiatric condition was 27%. Table 1 also describes the mean costs of mental health-related service use in the past year, by psychiatric trajectory (from no diagnosis to persistent psychiatric diagnosis). Bivariate analyses showed a non-significant association between psychiatric trajectory and mean annual costs. 

Comment 15: The 12-month service use and service use cost means are presented in Tables 1 and 2. Repeating the same findings across two tables should be avoided

Response: We have deleted the last line of Table 2 (overall services cost).

Comment 16: The paragraph on page 12 lacks a description of the cost associated with each service? For e.g., although CAPS is not a highly prevalent service, the associated cost makes for a lion contribution to the public purse. This needs to be presented and discussed.

Response: Thank you very much for your suggestion. We edited this paragraph as follows:

Utilisation of mental health services in the previous 12 months and associated cost by type of service are presented in Table 2. Overall, 10.0% of the sample (n=143) used some sort of health, education, criminal justice or social care service for mental health problems. Disaggregating by sectors, the health sector had highest proportion of service users (9%), while the education and social care and criminal justice sectors were less frequently contacted with a 1.8% and 1.3% of users, respectively. Outpatient mental health services, most notably psychologists and psychiatrists in settings other than community mental health clinics, were the most frequently used services/professionals. Inpatient services and general health services such as GP/family doctor, paediatrician and emergency department, were less frequently used. 

In the education sector, school assistant was the most type of service used by young people, while guardianship council was the most frequently social care service contacted. The total cost of 12-month mental health-related service use for the public purse was 70,110.23 USD. The sector that presented higher total annual cost was the health sector, followed by the education and finally the social care and criminal justice sectors. The services that generated the greatest total costs for the heath sector were psychologist (11,339.64 USD) and CAPS (9,628.01 USD). Among those who used services, the average annual cost of service use amounted to 527.14 USD (SD= 908.10 USD, range=8.77- 7,605.58 USD, median=221.10 USD, interquartile range=545.28) per user. Individuals using CAPS (specialty mental health) services (1.1% of the sample) had the highest mean number of visits during the previous year and the highest associated costs among health services. The second highest mean costs in the health sector were related to hospitalizations in psychiatric hospitals and alcohol and drugs clinics, while the lowest mean costs were attributed to emergency department, paediatrician, outpatient alcohol and drugs and GP/family doctor contacts. Although only 0.1% of individuals used shelters, this type of social service had the highest associated mean cost. Education services were used by 1.8% of individuals and these services had the second highest associated mean costs. 

Comment 17: Discussion: In line 288, the use of the terms 'above and beyond' doesn't convey much. To the best of my knowledge, the current analysis nowhere helps to reach this conclusion of above and beyond. I am requesting authors to look at terminology closely.

Response: Thanks for your suggestion. We have removed this language.

We found that impact of mental health problems on daily life and parental stigma were the most consistent and robust drivers of mental health service use and associated costs..

Comment 18: Discussion: Lines 301-312 can be streamlined and better organised.

Response: We have edited the cited lines as follows: 

We did not find any study exploring the impact of parental stigmatising attitudes toward mental illness on child treatment costs. Other research has shown that parental stigma can impede problem recognition and help-seeking [17,43]. Higher stigma amongst parents and caregivers may discourage or delay service access for their children [16], which may reduce the short-term public sector direct costs of treatment but be detrimental in the long run. Future research needs to further explore the mechanisms through which parental stigma may be related to service/treatment selection and treatment adherence, in order to explain its impact on treatment costs. Moreover, as lower parental stigma may facilitate earlier service contact, it would be interesting to investigate if lower parental stigma may result in lower costs in the longer term. 

Comment 19: Discussion: Line 327: The number of CAPS users was less, but the number of visits and costs for those who used it were very high. These were not reflected in the discussion, nor were its implication for the restructuring health system.

Response: We appreciate your comment. We have edited the discussion as suggested:

In Brazil, access to CAPS does not require any referral. However, the number of CAPS services are limited, and they are focused on treatment of severe mental disorders [24]. The high costs incurred by the mental health sector for the treatment of psychiatric disorders in CAPS may be a result of both, the severity of patients consulting these services and because these services provide intensive outpatient treatments (reflected by the highest number of visits we found), which is costly compared with no-specialized services. It is important to highlight that the lack of youth-oriented primary care mental health services in Brazil which limits access to treatment. This could explain why we found low frequency of mental health-related contacts with GP/ family doctors. As a result, contact with specialist mental health services only happens when the disorder has significant negative impact on the lives of young people. (Moved from the conclusion as suggested in your last comment). In this sense, the organisation of a mental health network of care for adolescents, integrating primary care, social care, education, criminal justice and community youth-specialist services, according to the impact of cases, must be considered in Brazil to adequately plan and allocate scarce public budgets [47].

Comment 20: Discussion: The hypothesis stated that researchers were interested in examining the impact of persistence of psychiatric disorders from childhood to adolescence on service costs; however, the discussion did not give much attention to this part.

Response: Thanks for rising this important comment. We have included the following paragraph:

Contrary to what we expected, we did not find an association between disorder persistence and costs. Our analyses instead found that impact of the disorder on adolescent’s life was the most important clinical predictor and that this was what seemed to drive service use rather than type or persistence of diagnosis. Nevertheless, it is important to consider that we have estimated annual costs, and these do not necessarily reflect the cumulative economic costs of persistent cases across childhood and adolescence. 

Comment 21: Conclusion: Some of the text in the last paragraph of the conclusion, i.e. those referring to implications, can be moved to discussion and expanded further.

I'm not able to comment on cost analysis as this is not my area of expertise.

Response: Thanks for your suggestions, we have moved some conclusions to the discussion as explained in response to your Comment #20.

---

## [Decision Letter · Decision Letter 1]

25 Feb 2022

PONE-D-21-10663R1Utilisation and costs of mental health-related service use among adolescentsPLOS ONE

Dear Dr. Evans-Lacko,

Thank you for submitting your manuscript to PLOS ONE. There are still a few minor points that need addressing. Therefore, we invite you to submit a revised version of the manuscript that addresses the points raised during the review process. Please submit your revised manuscript by Apr 11 2022 11:59PM. If you will need more time than this to complete your revisions, please reply to this message or contact the journal office at plosone@plos.org. Please include the following items when submitting your revised manuscript:A rebuttal letter that responds to each point raised by the academic editor and reviewer(s). You should upload this letter as a separate file labeled 'Response to Reviewers'.A marked-up copy of your manuscript that highlights changes made to the original version. You should upload this as a separate file labeled 'Revised Manuscript with Track Changes'.An unmarked version of your revised paper without tracked changes. You should upload this as a separate file labeled 'Manuscript'.If applicable, we recommend that you deposit your laboratory protocols in protocols.io to enhance the reproducibility of your results. Protocols.io assigns your protocol its own identifier (DOI) so that it can be cited independently in the future. For instructions see: https://journals.plos.org/plosone/s/submission-guidelines#loc-laboratory-protocols. Additionally, PLOS ONE offers an option for publishing peer-reviewed Lab Protocol articles, which describe protocols hosted on protocols.io. Read more information on sharing protocols at https://plos.org/protocols?utm_medium=editorial-email&utm_source=authorletters&utm_campaign=protocols.

We look forward to receiving your revised manuscript.

Kind regards,

Therese van Amelsvoort

Academic Editor

PLOS ONE

Journal Requirements:

Reviewers' comments:

Reviewer's Responses to Questions

**Comments to the Author**

1. If the authors have adequately addressed your comments raised in a previous round of review and you feel that this manuscript is now acceptable for publication, you may indicate that here to bypass the “Comments to the Author” section, enter your conflict of interest statement in the “Confidential to Editor” section, and submit your "Accept" recommendation.

Reviewer #1: All comments have been addressed

Reviewer #2: (No Response)

2. Is the manuscript technically sound, and do the data support the conclusions?

Reviewer #1: (No Response)

Reviewer #2: Partly

3. Has the statistical analysis been performed appropriately and rigorously? 

Reviewer #1: (No Response)

Reviewer #2: I Don't Know

4. Have the authors made all data underlying the findings in their manuscript fully available?

Reviewer #1: (No Response)

Reviewer #2: Yes

5. Is the manuscript presented in an intelligible fashion and written in standard English?

Reviewer #1: (No Response)

Reviewer #2: Yes

6. Review Comments to the Author

Reviewer #1: (No Response)

Reviewer #2: It's good to see the quality of the manuscript and the text organisation has been improved. I'm sharing a few follow-up queries. A sincere request to authors to include line number where revisions are reflected on the non-tracked version. Without line numbers, its difficult to locate revisions.

Introduction:

Follow-up query to comment 3: In lines 57-59, the sentence's meaning is still not very clear. Are the mentioned demographic and clinical features associated with the frequent use of these services or the number of services used? Also, are these findings coming from adult literature, youth literature or across the age span?

Follow-up query to comment 4: In lines 65-69, the revised reasoning looks circular. The authors say that stigmatizing attitude among parents was associated with poor help-seeking. But in the following line states that little is known about the impact of parental stigma on help-seeking. Are authors implying that what is known about children is not applicable to adolescents? It's unclear how the existing findings relate to the gap that the author asserts in this para.

The organization of the text in the introduction section can be improved as a couple of paragraphs have just one or two lines.

Methods

In line 121- is this '-' a typo?

In Line 116-121, it is not clear to the reader how the study sample was reduced to 1881 from the original 2251. I'm guessing the authors included a sub-group of individuals meeting the age criteria. This needs to be clearly defined in the manuscript.

Results

Follow-up query to comment 14: It was helpful to see the description of the results in text. A few comments on the revised text: In line 253, the value in the text doesn't match with the value given in the table. If the rounding up was done, the number of decimal places should be consistent across the text. In line 262, there is no description of whether this service use is significantly different from other trajectories?

Follow-up query to comment 16: Again, it was helpful to see the description of the results in text. A few comments on the revised text: In line 276, there should be a full stop before starting the following sentence. Also, it may be good to start as "within the health sector, the outpatient…", so the reader is clear that now you are looking at the frequency of usage within each sector.

In line 289, it might be good to specify the mean visit per person, so it's clear to the reader that the mean refers to the mean visit per person.

The source of data in lines 334-337 is not provided.

There are a lot of supplementary data files, but except for two supplementary files, others were not referenced or explained in the results section. It would be helpful to understand the reason for including all this analysis but not using them in results.

Discussion

The findings discussed in lines 342-346 don't match the result. The service use was different among those with different trajectories. However, the costs were not different among these groups. Does any of the existing analysis carried out by the author can explain why the difference in frequency of use didn't result in differences in cost between these groups.

Follow-up query to comment 18: The revision looks good. However, in line 347, the authors' assertion that these findings are novel seems misleading. As in quick succession, they have indicated that other studies have also found similar findings.

Follow-up query to comment 19: The discussion points on CAPS results in lines 380-385 are difficult to follow. There is no mention in results about the severity of disorders and types of service accessed. So understanding this link in discussion is difficult.

7. PLOS authors have the option to publish the peer review history of their article (what does this mean?). If published, this will include your full peer review and any attached files.

Reviewer #1: No

Reviewer #2: **Yes: **Kanika Malik

---

## [Author Response · Author response to Decision Letter 1]

22 Apr 2022

We appreciate the careful revision of our manuscript and the comments of the reviewers. We are pleased to be invited to submit the revised version of our paper to PLOS ONE. 

Please find attached both an unmarked version of the revised manuscript and one version with changes marked in red. Our point-by-point responses to the reviewers’ comments (unquoted italics) and details of the changes we have performed to our revised manuscript are given below. 

Reviewer’s general comment: It's good to see the quality of the manuscript and the text organisation has been improved. I'm sharing a few follow-up queries. A sincere request to authors to include line number where revisions are reflected on the non-tracked version. Without line numbers, it’s difficult to locate revisions.

We appreciate your careful revision of our manuscript and all your comments and suggestions. We have included the line number to locate the revisions of our manuscript. 

Follow-up query to comment 3: In lines 57-59, the sentence's meaning is still not very clear. Are the mentioned demographic and clinical features associated with the frequent use of these services or the number of services used? Also, are these findings coming from adult literature, youth literature or across the age span?

Thanks for your comment. We have edited this paragraph as follows: 

Some studies from high–income countries about mental health-related service use among young people suggest that lower socioeconomic status, as well as clinical features (illness severity and impact of disorders) increase the likelihood of health, special education, and social care services, while male gender and older age are associated with more criminal justice services contacts [11,13,14]. (Introduction, lines 58-60)

Follow-up query to comment 4: In lines 65-69, the revised reasoning looks circular. The authors say that stigmatizing attitude among parents was associated with poor help-seeking. But in the following line states that little is known about the impact of parental stigma on help-seeking. Are authors implying that what is known about children is not applicable to adolescents? It's unclear how the existing findings relate to the gap that the author asserts in this para.

Thanks for your comment. We have asserted this paragraph in order to explain that there is a clear link established between stigma influencing help-seeking among adults and adolescents (we have added one reference on the later topic), however, much less data on parental stigma in relation to their child’s service use has been published. We have added the only study that investigated the association between parental stigma and young people’s service use in the UK, published on January 27th, 2022. This paragraph was edited as follows:

Stigma can also influence help-seeking. Research in adult populations show clear links between stigma and reduced help-seeking [16], reduced adherence to treatment and early withdrawal from services [16,17]. Additionally, stigma is one of the most reported barriers to help-seeking amongst adolescents [18]. Families also play a central role in young people’s contact with services. However, there is limited research about how parental stigma could impact on service use and costs. Stigmatising attitudes toward mental illness amongst parents may influence service contacts due to shame and fear of labelling their youth’s mental health condition [19]. We know of two papers from the UK (using the same data), where lower mental illness-related stigma among caregivers (n=407), was associated with an increased likelihood of young people’s mental health service use [19], though it was not associated with costs [20]. We know of no studies on the association between mental illness-related stigma among caregivers and young people’s service use and costs outside the UK or in low and middle-income countries (LMICs), where families and young people may face different types of barriers to mental health care. (Introduction, lines 64-77)

New References were updated:

18. Aguirre Velasco A, Cruz ISS, Billings J, Jimenez M, Rowe S. What are the barriers, facilitators and interventions targeting help-seeking behaviours for common mental health problems in adolescents? A systematic review. BMC Psychiatry. 2020;20: 293. doi:10.1186/s12888-020-02659-0

20. Ribeiro WS, Romeo R, King D, Owens S, Gronholm PC, Fisher HL, et al. Influence of stigma, sociodemographic and clinical characteristics on mental health-related service use and associated costs among young people in the United Kingdom. Eur Child Adolesc Psychiatry. 2022. doi:10.1007/s00787-022-01947-2

The organization of the text in the introduction section can be improved as a couple of paragraphs have just one or two lines.

Thanks for your suggestion. We have merged two paragraphs: lines 77 and 92. 

Methods

In line 121- is this '-' a typo?

We have edited this sentence as follows:

Among those contacted, 1,400 (74.4%) guardians (in 93.1% of cases the biological mother) completed the interview, 982 (70.1%) by telephone and 418 (29.9%) face-to-face (See flow chart in S1 Fig.). (Line 126)

In Line 116-121, it is not clear to the reader how the study sample was reduced to 1881 from the original 2251. I'm guessing the authors included a sub-group of individuals meeting the age criteria. This needs to be clearly defined in the manuscript.

We appreciate your comment. The Brazilian High-Risk Cohort (BHRC) comprises 2511 children, 2010 of them (80%) participated at the first follow-up. Due to an administrative error, we were only able to contact a subsample of those who participated at first follow-up (94%, n=1881) to participate in our additional research and thus we invited 1881 guardians to complete further questions on mental health-related service use and associated barriers to care. Of those participants who we contacted, 1,400 (74%) agreed to participate in our study. 

We have included the following changes (in red) in lines 119-128:

Cohort participants were interviewed at baseline (aged 6-14 years, calendar year: 2010-2011, n=2,511), and at first follow-up (n=2010, aged 9-17 years, calendar year 2014, 80% retention rate). Due to an administrative error, we were only able to contact a subsample of those who participated at the first follow-up (94%, n=1,881) to respond to a supplementary interview which included a comprehensive assessment of mental health-related service use (calendar year: 2014-2015, young people participants aged 10-18 years). Among those contacted, 1,400 (74.4%) guardians (in 93.1% of cases the biological mother) completed the interview, 982 (70.1%) by telephone and 418 (29.9%) face-to-face (See flow chart in S1 Fig.). There were no significant differences in persistence of psychopathology or impact of psychopathology on adolescents’ lives among respondents versus non-respondents. 

Results

Follow-up query to comment 14: It was helpful to see the description of the results in text. A few comments on the revised text: In line 253, the value in the text doesn't match with the value given in the table. If the rounding up was done, the number of decimal places should be consistent across the text. In line 262, there is no description of whether this service use is significantly different from other trajectories?

We appreciate your comment. We have edited the value in line 259 (10.6%) and we have included the description of whether the service use was significantly different by trajectories:

Unadjusted odds ratios of any service use among participants with persistent diagnosis were 7.14 (95%CI=4.50-11.58, p<0.001) compared with participants with no diagnosis, OR=1.72 (95%CI=1.02-2.91, p=0.043) compared with incident and OR=2.62 (95%CI=1.52-4.49, p<0.001) with remittent diagnosis. (Lines 267-270).

Follow-up query to comment 16: Again, it was helpful to see the description of the results in text. A few comments on the revised text: In line 276, there should be a full stop before starting the following sentence. Also, it may be good to start as "within the health sector, the outpatient…", so the reader is clear that now you are looking at the frequency of usage within each sector.

Thank you very much for your suggestion, we have corrected this as requested. (Line 284)

In line 289, it might be good to specify the mean visit per person, so it's clear to the reader that the mean refers to the mean visit per person.

Thank you very much for your suggestion, we have corrected this as requested. (Line 297)

The source of data in lines 334-337 is not provided.

We appreciate your comment. Due to lines 334-337 are part of the first paragraph of the discussion, we wonder whether you are referring to lines 324-327. If so, we have edited this sentence as follows:

When looking at predictors of costs according to sector (Table 4), disorder impact was associated with greater health sector service use (predicted mean cost by each impact score= 66.26 USD, p=0.019). (Lines 343-345)

There are a lot of supplementary data files, but except for two supplementary files, others were not referenced or explained in the results section. It would be helpful to understand the reason for including all this analysis but not using them in results.

Supplementary analyses were explained in the methods section. Following your suggestion, we have included the description of supplementary tables in the results sections as follows:

Regarding to service use, lines 312-326: 

Having an incident, remittent or persistent psychiatric disorder, as well as the higher impact of behavioural and emotional difficulties on the adolescents’ lives and lower parental stigma, all predicted higher odds of any 12-month service use in unadjusted analyses (S2 Table), in models adjusted by sociodemographic characteristics (S4-S6 Tables) and multivariable analyses (Table 3). Service contacts in the health sector were also predicted by the same factors. Service use in the educational sector was predicted by diagnosis trajectory, impact and lower stigma in the unadjusted analyses (S2 Table), and in the models adjusted by sociodemographic characteristics (S4-S6 Tables). However, disorder persistence did not remain significant in multivariable analyses, where impact, lower parental stigma and low SEG showed significant association with education service use (Table 3). Although persistence of the disorder and higher impact of behavioural and emotional difficulties on the adolescents’ lives were associated with social care and criminal justice service use in the unadjusted analyses (S2 Table), and in the models adjusted by sociodemographic characteristics (S4-S5 Table), there were no factors significantly associated with use of social care and criminal justice services in multivariable analyses (Table 3). 

Additionally, to clarify that adjusted estimates are showed in Table 3, we have changed AOR (adjusted odds rations) instead OR (Line 328). 

Regarding to services costs, lines 334-338:

When all three sectors were combined into a single total cost variable, greater impact and lower parental stigma were associated with higher costs in unadjusted analyses (S3 Table), in the models adjusted by sociodemographic characteristics (S8-S9 Tables) and multivariable analyses (Table 4). White ethnicity was also associated with higher costs in multivariable analyses (Table 4)….

Lines 345-350:

We did not find any significant association of psychiatric trajectories, impact of disorder or parental stigma with education sector’s costs (S3 Table, S7-S9 Tables, and Table 4). Although persistence of the disorder was associated with social care and criminal justice service’s costs in the unadjusted analyses (S3 Table), and in the models adjusted by sociodemographic characteristics (S7 Table), we did not find any significant factors associated with social care/criminal justice sectors’ costs in multivariable analyses (Table 4).

Discussion

The findings discussed in lines 342-346 don't match the result. The service use was different among those with different trajectories. However, the costs were not different among these groups. Does any of the existing analysis carried out by the author can explain why the difference in frequency of use didn't result in differences in cost between these groups.

Thanks for your comment. Apologise for the confusion. We have corrected that we were discussing in this part the results of service costs instead of service use, as follows:

Our analyses instead found that impact of the disorder on adolescent’s life was the most important clinical predictor and that this was what seemed to drive service costs rather than type or persistence of diagnosis. (Lines 366-368)

Regarding to your question about whether our analyses can explain why the difference in frequency of use did not result in differences in cost between these groups, service costs reflect the intensity of care (number of visits, as explained in the introduction) while service use reflect the probability of having any contact with the services. In this sense, our results suggest that the persistence of a diagnosis can increase the likelihood of service use but does not necessarily have an impact on the number of contacts per year. Unfortunately, as pointed out in the limitations of our study (lines 433-435), the scarce number of service users by specific services did not allow us to perform analyses to compare, for example, whether the trajectory it is associated to the selection of CAPS (high-costs services) instead primary care (low-costs services) services contacts. We have added the following sentence to the discussion:

This finding suggests that although the trajectory of the diagnosis is associated with a greater likelihood of having any contact with mental health-related services, this association did not translate to costs which also reflects the intensity and / or type of care and number of visits per year. Nevertheless, it is important to consider that we have estimated annual costs, and these do not necessarily reflect the cumulative economic costs of persistent cases across childhood and adolescence. (Lines 368-374).

Follow-up query to comment 18: The revision looks good. However, in line 347, the authors' assertion that these findings are novel seems misleading. As in quick succession, they have indicated that other studies have also found similar findings.

Thanks for your comment. We have edited this sentence as follows: 

We also found that lower parental stigma was associated with greater service use and higher costs. (Line 371)

Additionally, as we are aware of a new paper (published on January 27th, 2022) that investigate the association between parental stigma and service use costs, we have edited the paragraph in lines (381-384):

We only found one study [20] exploring the impact of parental stigmatising attitudes toward mental illness on child service use costs. The cited study, conducted in the UK, with a smaller sample size, did not find an association between parental stigma and young people’s service use costs.

Follow-up query to comment 19: The discussion points on CAPS results in lines 380-385 are difficult to follow. There is no mention in results about the severity of disorders and types of service accessed. So understanding this link in discussion is difficult.

We appreciate your comment. We eliminated the sentence that connected severity of cases with CAPS’s costs and the sentence in the revised version appears as follows:

The high costs incurred by the mental health sector for the treatment of psychiatric disorders in CAPS may be because these services provide intensive outpatient treatments (reflected by the highest number of visits we found), which is costly compared with no-specialized services. (Lines 408-411)

---

## [Decision Letter · Decision Letter 2]

25 May 2022

PONE-D-21-10663R2Utilisation and costs of mental health-related service use among adolescentsPLOS ONE

Dear Dr. Lacko,

Thank you for submitting your manuscript to PLOS ONE. There are still some minor questions remaining before we can accept your manuscript.

We look forward to receiving your revised manuscript.

Kind regards,

Therese van Amelsvoort

Academic Editor

PLOS ONE

Journal Requirements:

Reviewers' comments:

Reviewer's Responses to Questions

**Comments to the Author**

1. If the authors have adequately addressed your comments raised in a previous round of review and you feel that this manuscript is now acceptable for publication, you may indicate that here to bypass the “Comments to the Author” section, enter your conflict of interest statement in the “Confidential to Editor” section, and submit your "Accept" recommendation.

Reviewer #1: All comments have been addressed

Reviewer #2: (No Response)

2. Is the manuscript technically sound, and do the data support the conclusions?

Reviewer #1: (No Response)

Reviewer #2: Partly

3. Has the statistical analysis been performed appropriately and rigorously? 

Reviewer #1: (No Response)

Reviewer #2: Yes

4. Have the authors made all data underlying the findings in their manuscript fully available?

Reviewer #1: (No Response)

Reviewer #2: Yes

5. Is the manuscript presented in an intelligible fashion and written in standard English?

Reviewer #1: (No Response)

Reviewer #2: Yes

6. Review Comments to the Author

Reviewer #1: My comments were already addressed, to my satisfaction, in the previous version of the manuscript.

Reviewer #2: Thank you for addressing most of my comments. The manuscript is looking well. Here are a suggested few minor revisions:

In lines 59-61, it is still unclear whether these features increase the likelihood of service usage or the number of services used.

In lines 121-122, the authors have stated that 'due to an administrative error, we were only able to contact a subsample of those who participated at the first follow-up'. It will be good to have more details about this error in the study limitation and how this may have affected the validity of the results.

For lines 261-272, I'm really struggling to figure out where all these percentage values, OR, and RR are coming from, as they are not given in Table 1. If it's from some supplementary table, please cite it.

In lines 258 and 267, the value in the text doesn't match the value given in the table. If the rounding up was done, the number of decimal places should be consistent across the text

7. PLOS authors have the option to publish the peer review history of their article (what does this mean?). If published, this will include your full peer review and any attached files.

Reviewer #1: No

Reviewer #2: **Yes: **KANIKA MALIK

---

## [Author Response · Author response to Decision Letter 2]

17 Jun 2022

We appreciate the careful revision of our manuscript and the comments of the reviewers. We are pleased to be invited to submit the revised version of our paper to PLOS ONE. 

Please find attached both an unmarked version of the revised manuscript and one version with changes marked in red. Our point-by-point responses to the reviewers’ comments (unquoted italics) and details of the changes we have performed to our revised manuscript are given below. 

Reviewer #1: My comments were already addressed, to my satisfaction, in the previous version of the manuscript.

Many thanks for your revision and we are glad of having addressed your comments appropriately.

Reviewer #2: Thank you for addressing most of my comments. The manuscript is looking well. Here are a suggested few minor revisions:

In lines 59-61, it is still unclear whether these features increase the likelihood of service usage or the number of services used.

Thanks for your suggestion. We have edited this paragraph as follows:

Lines 59-62: Some studies from high–income countries about mental health-related service use among young people suggest that lower socioeconomic status, as well as clinical features (illness severity and impact of disorders) increase the likelihood of service usage in the health, special education, and social care sectors, while male gender and older age are associated with more criminal justice services contacts [11,13,14].

In lines 121-122, the authors have stated that 'due to an administrative error, we were only able to contact a subsample of those who participated at the first follow-up'. It will be good to have more details about this error in the study limitation and how this may have affected the validity of the results.

We appreciate your comment. We have provided more details about this error in lines 121-123 (Methods section):

Due to an administrative error, the service use questions were not included in the interview schedule for the first 129 participants (6%) of the BHRC first follow-up. Therefore, we were only able to contact a subsample of those who participated at the first follow-up (94%, n=1,881) to respond to a supplementary interview which included a comprehensive assessment of mental health-related service use (calendar year: 2014-2015, young people participants aged 10-18 years).

And we added in the Limitations subheading the following sentences:

Lines 440-442: Fifth, due to an administrative error we were unable to contact the first 6% of first follow-up BHRC participants. This could reduce our sample size, but as this was a random error, we do not believe that it affected the results.

For lines 261-272, I'm really struggling to figure out where all these percentage values, OR, and RR are coming from, as they are not given in Table 1. If it's from some supplementary table, please cite it.

We appreciate the comment of the reviewer. 

To clarify the percentage of service utilization we edited the text as follows: 

Lines 261-267: 23.3% (n= 260) of adolescents had a psychiatric disorder in the previous 12 months, of which (54.3%) 261 were incident and 149 (45.7%) persistent cases since baseline. 213 (15.2%) participants had remitted from a baseline psychiatric diagnosis. [We moved the next sentence here] 73 (22.4%) of those who presented with a psychiatric disorder (32 incident and 41 persistent) reported using some type of service for their mental health in the previous twelve months. The proportion of service use among those who presented a persistent psychiatric condition was 27.5%.

Regarding to the following text added in our second revision:

Lines 267-270: Unadjusted odds ratios of any service use among participants with persistent diagnosis were 7.22 [we amended these decimals] (95%CI=4.50-11.58, p<0.001) compared with participants with no diagnosis, OR=1.72 (95%CI=1.02-2.91, p=0.043) compared with incident and OR=2.62 (95%CI=1.52-4.49, p<0.001) with remittent diagnosis.

These results were added with the aim of addressing your comment on whether there were differences in service use between trajectories. In Supplementary Table 2 we presented these differences taking as reference ‘no diagnosis’, and the results presented in this text (Lines 267-270) would be taking as reference ‘persistent psychiatric conditions’. These results were not presented in a specific table. 

Regarding to other statistics informed in the lines cited by the reviewer, Table 1 describes sociodemographic and clinical characteristics of participants, according to mental health conditions trajectories. We added only in the text (i.e. not in a specific table) the results of bivariate statistics for disorder impact (one of our main predictors) and type of broad diagnostic group by trajectories. To clarify this, we decided to only maintain the mean differences in disorder impact by trajectories and we removed the differences in terms of type of diagnostic group, as this was not the focus of our study. Changes were added as follows: 

Lines 271-275: Table 1 present mean disorder impact by psychiatric trajectories. Unadjusted generalised regression models showed that persistent cases presented greater mean difference in disorder impact (SDQ scores) by 2.34, (95%CI=2.11-2.58, p<0.001) compared with no diagnosis, 1.14 (95% CI=0.85-1.42, p<0.001), compared with incident cases, and 1.84 (95% CI=1.56-2.12, p<0.001), compared with remittent cases. 

In lines 258 and 267, the value in the text doesn't match the value given in the table. If the rounding up was done, the number of decimal places should be consistent across the text

Thanks for your suggestion, we amended these values as follows:

Line 260

14.51 years (s.d=1.98).

Line 267

27.5%.

And we also corrected the decimal places in the percentage of overall service use in Line 285 and Table 1

10.2%

---

## [Decision Letter · Decision Letter 3]

12 Aug 2022

Utilisation and costs of mental health-related service use among adolescents

PONE-D-21-10663R3

Dear Dr. Evans-Lacko,

We’re pleased to inform you that your manuscript has been judged scientifically suitable for publication and will be formally accepted for publication once it meets all outstanding technical requirements.

Kind regards,

Dylan A Mordaunt, MD, MPH, FRACP

Academic Editor

PLOS ONE

Additional Editor Comments (optional):

Thank you for your resubmission. This now meets the criteria for publication.

Reviewers' comments:

Reviewer's Responses to Questions

**Comments to the Author**

1. If the authors have adequately addressed your comments raised in a previous round of review and you feel that this manuscript is now acceptable for publication, you may indicate that here to bypass the “Comments to the Author” section, enter your conflict of interest statement in the “Confidential to Editor” section, and submit your "Accept" recommendation.

Reviewer #1: All comments have been addressed

2. Is the manuscript technically sound, and do the data support the conclusions?

Reviewer #1: (No Response)

3. Has the statistical analysis been performed appropriately and rigorously? 

Reviewer #1: (No Response)

4. Have the authors made all data underlying the findings in their manuscript fully available?

Reviewer #1: (No Response)

5. Is the manuscript presented in an intelligible fashion and written in standard English?

Reviewer #1: (No Response)

6. Review Comments to the Author

Reviewer #1: My comments were already addressed, to my satisfaction, in the previous version of the manuscript.

7. PLOS authors have the option to publish the peer review history of their article (what does this mean?). If published, this will include your full peer review and any attached files.

Reviewer #1: No

---

## [Editor Report · Acceptance letter]

1 Sep 2022

PONE-D-21-10663R3 

Utilisation and costs of mental health-related service use among adolescents 

Dear Dr. Evans-Lacko:

I'm pleased to inform you that your manuscript has been deemed suitable for publication in PLOS ONE. Congratulations! Your manuscript is now with our production department. 

Kind regards, 

on behalf of

Associate Professor Dylan A Mordaunt 

Academic Editor

PLOS ONE